# Fast Monte Carlo Tree Diffusion:
# 100× Speedup via Parallel Sparse Planning

**Jaesik Yoon**\*
KAIST & SAP
jaesik.yoon@kaist.ac.kr

**Hyeonseo Cho**\*
KAIST
hyeonseo.cho@kaist.ac.kr

**Yoshua Bengio**
Mila – Quebec AI Institute
Université de Montréal
yoshua.bengio@mila.quebec

**Sungjin Ahn**
KAIST
sungjin.ahn@kaist.ac.kr

## Abstract

Diffusion models have recently emerged as a powerful approach for trajectory planning. However, their inherently non-sequential nature limits their effectiveness in long-horizon reasoning tasks at test time. The recently proposed Monte Carlo Tree Diffusion (MCTD) offers a promising solution by combining diffusion with tree-based search, achieving state-of-the-art performance on complex planning problems. Despite its strengths, our analysis shows that MCTD incurs substantial computational overhead due to the sequential nature of tree search and the cost of iterative denoising. To address this, we propose Fast-MCTD, a more efficient variant that preserves the strengths of MCTD while significantly improving its speed and scalability. Fast-MCTD integrates two techniques: Parallel MCTD, which enables parallel rollouts via delayed tree updates and redundancy-aware selection; and Sparse MCTD, which reduces rollout length through trajectory coarsening. Experiments show that Fast-MCTD achieves up to 100× speedup over standard MCTD while maintaining or improving planning performance. Remarkably, it even outperforms Diffuser in inference speed on some tasks, despite Diffuser requiring no search and yielding weaker solutions. These results position Fast-MCTD as a practical and scalable solution for diffusion-based inference-time reasoning.

## 1 Introduction

Diffusion models have recently emerged as a powerful paradigm for trajectory planning, leveraging offline datasets to generate complex, high-quality trajectories through iterative denoising [1, 2, 12, 36]. Unlike autoregressive planners that generate trajectories sequentially via forward dynamics [3, 10, 11], diffusion-based approaches like Diffuser [12] generate trajectories holistically, mitigating issues such as long-term dependencies and cumulative error.

Despite these advantages, diffusion planners often struggle with complex reasoning at test time, especially in long-horizon tasks. They may produce plausible but suboptimal trajectories that fail to accomplish complex goals. To address this, *inference-time scaling* has emerged as a promising approach to enhance reasoning by adding computational procedures—such as verification and refinement—during inference [16, 20, 21, 26, 35, 36].

---

\*Equal contribution. Correspondence to Jaesik Yoon and Sungjin Ahn <jaesik.yoon@kaist.ac.kr and sungjin.ahn@kaist.ac.kr>.

39th Conference on Neural Information Processing Systems (NeurIPS 2025).

However, designing a diffusion-based method that supports inference-time scaling remains challenging. This is primarily due to a fundamental mismatch: while reasoning often requires sequential or causal processing, diffusion models are inherently non-sequential and generate trajectories in a non-causal manner by design. As a result, simple strategies such as increasing the denoising depth or employing best-of-$N$ sampling have been shown to offer only limited improvements [33].

To address this challenge, Yoon et al. [33] proposed *Monte Carlo Tree Diffusion* (MCTD), which integrates diffusion-based planning with sequential search through tree-based reasoning, akin to Monte Carlo Tree Search (MCTS) [7]. MCTD adopts Diffusion Forcing [2] as the backbone and reinterprets its block-wise denoising as a causal tree rollout, introducing sequential structure while preserving the global generative strengths of diffusion models. By structuring trajectory generation as a tree search, MCTD enables systematic exploration and exploitation, helping the model escape local optima and discover higher-quality trajectories.

While MCTD has demonstrated impressive performance in complex, long-horizon tasks that conventional methods fail to solve, further analysis reveals that it suffers from significant computational inefficiencies due to two key bottlenecks: the sequential nature of MCTS, which updates tree statistics after each iteration, and the iterative denoising process inherent to diffusion models. Unfortunately, this inefficiency is most pronounced in the very long-horizon settings where MCTD's planning capabilities are most beneficial. Thus, improving the efficiency of MCTD is the most critical challenge for establishing it as a broadly practical solution for diffusion-based inference-time scaling.

In this paper, we propose Fast Monte Carlo Tree Diffusion (Fast-MCTD), a framework that significantly reduces the computational overhead of tree search and iterative denoising while preserving the strong planning capabilities of MCTD. Fast-MCTD integrates two key optimization techniques: Parallel MCTD (P-MCTD) and Sparse MCTD (S-MCTD). P-MCTD accelerates the tree search process by enabling parallel rollouts, deferring tree updates until multiple searches are completed, introducing redundancy-aware selection, and parallelizing both expansion and simulation steps. S-MCTD further improves efficiency by planning over coarsened trajectories, using diffusion models trained on these compressed representations. This not only reduces the cost of iterative denoising but also lowers overall search complexity by effectively shortening the planning horizon.

Experimental results show that Fast-MCTD achieves substantial *speedups—up to 100×* on some tasks—compared to standard MCTD, while maintaining comparable or superior planning performance. Remarkably, Fast-MCTD also outperforms Diffuser in inference speed on some tasks, despite Diffuser's lack of search and its substantially inferior performance.

The main contributions of this paper are twofold. First, we introduce Fast-MCTD, a framework that improves the efficiency of MCTD through parallelization and rollout sparsification. Second, we empirically show that Fast-MCTD achieves substantial speedups—*up to two orders of magnitude*—while maintaining strong planning performance, demonstrating its practical effectiveness in challenging long-horizon tasks.

## 2 Preliminaries

### 2.1 Diffusion models for planning

**Diffuser** [12] formulates planning as a generative denoising process over full trajectories, defined as

$$\mathbf{x} = \begin{bmatrix} s_0 & s_1 & \cdots & s_T \\ a_0 & a_1 & \cdots & a_T \end{bmatrix}, \tag{1}$$

where $T$ is the trajectory length and $(s_t, a_t)$ denotes the state and action at time $t$, respectively. During inference, trajectories are iteratively denoised from noisy samples, effectively reversing a forward diffusion process. Since the learned denoising model alone does not inherently optimize for returns or task objectives, Diffuser incorporates a guidance function $\mathcal{J}_\phi(\mathbf{x})$ inspired by classifier-guided diffusion [8]. Specifically, the guidance function biases trajectory generation towards high-return outcomes by modifying the sampling distribution as:

$$\tilde{p}_\theta(\mathbf{x}) \propto p_\theta(\mathbf{x}) \exp\left(\mathcal{J}_\phi(\mathbf{x})\right), \tag{2}$$

thus explicitly guiding the denoising process toward optimal trajectories at test time.

**Diffusion Forcing** [2] extends the Diffuser framework by introducing token-level denoising control within trajectories. Specifically, Diffusion Forcing tokenizes the trajectory $\mathbf{x}$, enabling different tokens to be denoised at distinct noise levels. This selective, partial denoising allows the model to generate only the tokens exhibiting higher uncertainty, such as future plan tokens.

## 2.2 MCTD: Monte Carlo Tree Diffusion

MCTD unifies tree search and diffusion-based planning by integrating three key concepts.

**(1) Denoising as tree rollout.** Unlike traditional MCTS, which expands trees over individual states, MCTD partitions a trajectory $\mathbf{x}$ into fixed-length subplans $\mathbf{x}_s$, $\mathbf{x} = (\mathbf{x}_1, \ldots, \mathbf{x}_S)$, each treated as a high-level node. It applies a semi-autoregressive denoising schedule using Diffusion Forcing, where earlier subplans are denoised faster, conditioning later ones on previous outputs, approximating:

$$p(\mathbf{x}) \approx \prod_{s=1}^{S} p(\mathbf{x}_s | \mathbf{x}_{1:s-1}) \qquad (3a) \qquad p(\mathbf{x}|\mathbf{g}) \approx \prod_{s=1}^{S} p(\mathbf{x}_s | \mathbf{x}_{1:s-1}, g_s) \qquad (3b)$$

where $g_s$ is a guidance level (weight) for the classifier-guided sampling [8] (Equation 2). This preserves the global coherence of diffusion models while enabling intermediate rollouts akin to MCTS, substantially reducing tree depth.

**(2) Guidance levels as meta-actions.** To address exploration-exploitation trade-offs in large or continuous action spaces, MCTD introduces meta-actions via guidance schedules. A guidance schedule $\mathbf{g} = (g_1, \ldots, g_S)$ assigns discrete control modes—e.g., GUIDE or NO_GUIDE—to each subplan, modulating whether it is sampled from an unconditional prior (Equation 3a) or a guided distribution (Equation 3b). This allows selective guidance within a single diffusion process, enabling flexible trade-off control.

**(3) Jumpy denoising as fast simulation.** For efficient simulation, MCTD uses jumpy denoising (e.g., Denoising Diffusion Implicit Models (DDIM) [28]) to skip denoising steps and rapidly complete remaining trajectory segments. Given subplans $\mathbf{x}_{1:s}$, the rest $\tilde{\mathbf{x}}_{s+1:S}$ is coarsely denoised as: $\tilde{\mathbf{x}}_{s+1:S} \sim p(\mathbf{x}_{s+1:S} | \mathbf{x}_{1:s}, \mathbf{g})$, yielding a full trajectory $\tilde{\mathbf{x}}$ for evaluation.

**The four steps of an MCTD round.** These components are instantiated through the four canonical steps of MCTS adapted to operate over subplans within a diffusion-based planning framework. See [33] for a detailed pseudocode of the MCTD method.

1. **Selection.** MCTD traverses the tree from the root to a leaf using a selection strategy like Upper Confidence Boundary of Tree (UCT) [14]. Each node corresponds to a temporally extended subplan, not a single state, reducing tree depth and improving abstraction. The guidance schedule $\mathbf{g}$ is updated during traversal to balance exploration (NO_GUIDE) and exploitation (GUIDE).

2. **Expansion.** Upon reaching a leaf, a new subplan $\mathbf{x}_s$ is sampled using the diffusion forcing, conditioned on the chosen meta-action $g_s$. Sampling may follow an exploratory prior $p(\mathbf{x}_s | \mathbf{x}_{1:s-1})$ or a goal-directed distribution $p(\mathbf{x}_s | \mathbf{x}_{1:s-1}, g_s)$.

3. **Simulation.** Fast jumpy denoising (e.g., via DDIM) completes the remaining trajectory $\tilde{\mathbf{x}}$ after expansion. Although approximate, this is computationally efficient and sufficient for plan evaluation using a reward function $r(\tilde{\mathbf{x}})$.

4. **Backpropagation.** The obtained reward is backpropagated through the tree to update value estimates and refine guidance strategies for future planning rounds. This enables adaptive, reward-informed control of exploration and exploitation.

## 3 The planning horizon dilemma

Despite its strong performance on long-horizon and complex planning tasks, MCTD suffers from a fundamental computational inefficiency. This is because it requires repeated partial denoising

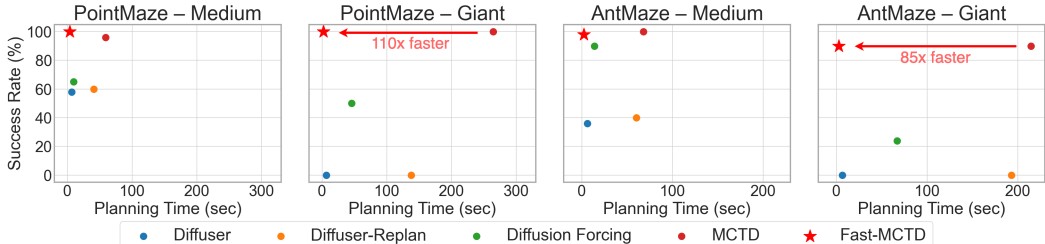

Figure 1: **Planning time vs. success rate.** As maze size increases, most diffusion-based planners degrade in performance. MCTD maintains high success rates but with long planning times, reflecting the Planning Horizon Dilemma—better performance requires longer planning time. Fast-MCTD breaks this trade-off, achieving strong performance with much faster planning.

operations to generate feasible subplans. Worse yet, as the planning horizon increases, the search space grows exponentially, resulting in significantly higher computational cost.

Ironically, the key strength of diffusion-based planning—its capacity for long-horizon reasoning—becomes a principal bottleneck when integrated with the inherently sequential nature of Monte Carlo Tree Search. In effect, the very scenarios where MCTD's planning capabilities are most advantageous are also those where it incurs the highest computational cost. We refer to this tension between the global trajectory reasoning of Diffuser and the step-by-step search process of MCTS as the *Planning Horizon Dilemma*.

More formally, consider a diffusion forcing schedule that generates a sequence of subplans. Let $N_{\text{child}}$ denote the branching factor at each tree node, corresponding to the cardinality of the meta-action space. Let $C_{\text{sub}}$ be the computational cost of performing partial denoising for a single subplan $\mathbf{x}_s$. Lastly, let $\bar{s} \leq S$ be the number of subplans required to reach the goal. Then, the total complexity of identifying a successful trajectory using MCTD is given by:

$$C_{\text{MCTD}} = \mathcal{O}(N_{\text{child}}^{\bar{s}} \cdot C_{\text{sub}}). \tag{4}$$

Although MCTD's exploration-exploitation strategy reduces average-case complexity, its computational cost still grows exponentially with the expected number of subplans $\bar{s}$.

This inefficiency of MCTD is clearly demonstrated in Figure 1, which compares planning time and success rates across two maze environments (point mass and ant robot) and two map sizes (medium and giant). As maze size increases from medium to giant, other diffusion-based planners such as Diffuser [12] and Diffusion Forcing [2] exhibit substantial drops in performance. In contrast, MCTD maintains a significant performance advantage due to its strong search capability. However, this advantage comes at a steep computational cost: in medium-sized mazes, MCTD requires roughly 8–10× more planning time than Diffuser, which further increases to 30–40× in giant-sized environments.

## 4 Fast Monte Carlo Tree Diffusion

The *Planning Horizon Dilemma* reveals two primary sources of inefficiency in MCTD: (1) **Between-rollout inefficiency** — partial denoising operations across different rollouts are executed serially, limiting parallelism; and (2) **Within-rollout inefficiency** — each rollout requires the Diffusion to process long trajectories, leading to substantial computational overhead. To address these challenges, we propose *Fast-MCTD*, which incorporates two key improvements: *Parallel Planning via P-MCTD*, to improve concurrency, and *Sparse Planning via S-MCTD*, to reduce the effective planning horizon. The overall processes are illustrated in Figure 2.

### 4.1 Parallel planning

**Independently parallel MCTD via delayed tree update.** We begin by parallelizing all four stages of MCTD through $K$ concurrent rollouts. In this setting, each iteration generates a *batch* of tree rollouts, each proceeding independently within the shared search tree. To avoid synchronization overhead, we adopt a *delayed tree update* strategy: all rollouts operate on a shared, fixed snapshot of

the tree, and updates to the tree (e.g., value estimates and visitation counts) are applied only after all rollouts in the batch are completed.

While this design allows for efficient parallel execution, it introduces a trade-off. As the batch size increases, the tree statistics used during search become increasingly stale, which can degrade planning performance by reducing the accuracy of selection and value propagation. To address this inefficiency, we propose a mitigation strategy in the following section. We further analyze this trade-off empirically in Section 6.4, identifying the optimal degree of parallelism that balances computational efficiency and search quality.

**Redundancy-Aware Selection.** While the delayed tree update enables parallel rollouts, it introduces a critical inefficiency: redundant node selection. That is, multiple rollouts within a batch independently select and thus may expand the same node, resulting in duplicated computation and limited exploration diversity. More specifically, this issue arises because the standard node selection policy in MCTD—Upper Confidence Bound for Trees (UCT) [14]—is designed for sequential search and does not account for concurrent expansions.

To address this issue, we introduce the *Redundancy-Aware Selection* (RAS) mechanism. Our key observation motivating this design is that the time spent on the Selection step, among the four stages of MCTD, is negligibly small: as shown in Table 1, it constitutes only 0.05% time of the MCTD steps. This allows us to remove the Selection (and the Backpropagation) step from the parallelization pipeline and instead implement it as a lightweight serial operation.

Table 1: Each step time of MCTD on PointMaze-Giant

| MCTD Step | Time (sec.) |
|---|---|
| Selection | 3e-4 (0.05%) |
| Expansion | **0.393 (70.9%)** |
| Simulation | **0.161 (29.0%)** |
| Backpropagation | 1e-4 (0.02%) |

Specifically, inspired by prior work on parallelized MCTS [17, 19, 25], we implement Redundancy-Aware Selection (RAS) by modifying the standard UCT-based selection criterion. During each parallel search phase, we temporarily incorporate an auxiliary visitation count variable $\hat{N}_i$, which *sequentially* tracks selections made within the current batch. This variable is reset to zero after the delayed tree update is applied. The resulting selection policy is defined as follows:

$$\pi(i) = \arg\max_{j \in \mathcal{C}(i)} \left( V_j + \beta \sqrt{\frac{\log(N_i + \hat{N}_i \cdot w)}{N_j + \hat{N}_j \cdot w}} \right), \tag{5}$$

where $\pi(i)$ denotes the node selection policy from node $i$, $\mathcal{C}(i)$ is the set of child nodes of node $i$, and $V_i$, $N_i$ represent the estimated value and visitation count of node $i$, respectively. The hyperparameter $w$ adjusts the exploration-exploitation balance for parallel search: when $w = 0$, the policy reduces to standard MCTD selection, and higher values of $w$ penalize nodes that have already been selected during the current batch, encouraging other rollouts to explore different parts of the tree.

**Parallel denoising on expansion and simulation.** Unlike conventional MCTS, where computational costs primarily stem from simulations, MCTD incurs substantial overhead in both expansion and simulation due to the expensive denoising operations. To enhance computational efficiency, we propose a unified batching strategy, termed *parallel denoising*, that simultaneously processes multiple subplans chosen by RAS during the expansion and simulation phases. Specifically, we implement a common `parallel_subplan` interface that batches denoising steps by scheduling noise levels and synchronizing DDIM [28] updates. To handle variable-length subplans and different guidance levels, subplans are zero-padded and packed into uniformly shaped tensors, enabling high-throughput parallel execution on GPUs.

### 4.2 Sparse planning

Another key contributor to the Planning Horizon Dilemma is the *within-rollout inefficiency*, wherein the cost of denoising increases with length of the rollout trajectories. This inefficiency persists even with the parallelization improvements introduced by P-MCTD. To address this issue, we propose *Sparse Monte Carlo Tree Diffusion (S-MCTD)*. The key idea is to incorporate trajectory coarsening—operating rollouts at a higher level of abstraction, as suggested in prior work [4, 5]—into the MCTD framework, thereby reducing the effective rollout length and overall computational cost.

Specifically, prior to training the diffusion model, we construct a dataset of coarse trajectories by subsampling every $H$ steps from the original trajectories, yielding $\mathbf{x}' = [x_1, x_{H+1}, x_{2H+1}, \dots]$.

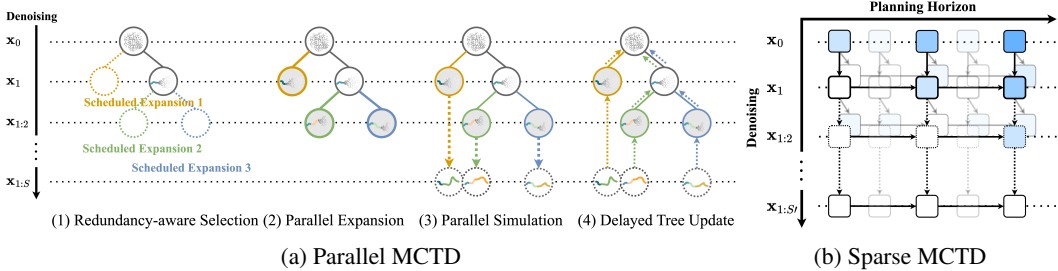

(a) Parallel MCTD

(b) Sparse MCTD

Figure 2: **Two key components of Fast-MCTD.** (a) **Parallel MCTD** accelerates planning by performing batched expansion and simulation on a partial denoising tree, followed by delayed tree updates. (b) **Sparse MCTD** reduces denoising overhead by planning over abstract sub-trajectories, significantly decreasing the number of subplans.

These coarse trajectories are then modeled using a specialized sparse diffusion planner. Owing to the shorter trajectory lengths, the computational cost of denoising each coarse subplan—denoted as $C_{\text{coarse}}$—is significantly lower than that of the original fine-grained subplans, $C_{\text{sub}}$. Furthermore, the overall search complexity is reduced, as planning now involves approximately $H$ times fewer subplans, i.e., $\mathbf{x}' = [\mathbf{x}'_1, \mathbf{x}'_2, \ldots, \mathbf{x}'_{S'}]$, where $S' \approx S/H$ given the original trajectory length $S$, as illustrated in Figure 2b. In terms of computational complexity, S-MCTD offers a substantial reduction compared to MCTD (Equation (4)):

$$C_{\text{S-MCTD}} = \mathcal{O}\left(N_{\text{child}}^{S/H} \cdot C_{\text{coarse}}\right) . \quad (6)$$

We observe that computational efficiency improves exponentially with the interval size $H$. However, excessively large intervals may overly abstract the planning task, potentially leading to degraded performance. We empirically investigate this trade-off in Section 6.4.

---

**Algorithm 1** Fast-MCTD

1: Initialize tree $\mathcal{T}$, selected node set $\mathcal{S}$
2: **while not** goal reached **do**
3:     Reset $\hat{N}_i$ for all node $i$ in $\mathcal{T}$
4:     **for** $i = 1$ to $K$ **do**
5:         Select node $v_i$ from $\mathcal{T}$ Equation (5)
6:         Update $\mathcal{S}$: $\mathcal{S} = \mathcal{S} \cup v_i$
7:     **end for**
8:     **for** $i = 1 \in \mathcal{S}$ **in parallel do**
9:         Expand node $v_i$ via `SparseExpand`
10:        Simulate with `SparseJumpySimulate`
11:     **end for**
12:     **for** $i = 1$ to $K$ **do**
13:         Backpropagate rewards through $\mathcal{T}$
14:     **end for**
15: **end while**
16: **return** best node from $\mathcal{T}$

---

Following prior work [4, 12, 33], we implement low-level control for executing each coarse subplan $\mathbf{x}'_s$ using heuristic controllers [12], value-based policies [4, 33], and inverse dynamics models [33].

### 4.3 Integrating Parallel and Sparse MCTD

Finally, we integrate the aforementioned Parallel MCTD (P-MCTD; Section 4.1) and Sparse MCTD (S-MCTD; Section 4.2) approaches into our final proposed method, termed Fast Monte Carlo Tree Diffusion (Fast-MCTD). We illustrate the details of Fast-MCTD in Algorithm 1.

## 5 Related work

Diffusion models [27] have demonstrated significant success in long-horizon trajectory planning, particularly in sparse-reward settings, by learning to generate plan holistically rather than forward modeling [1, 2, 12, 18, 36]. Given the computational intensity of the denoising process, many approaches have focused on improving efficiency through methods like hierarchical planning [5, 6, 9]. To address the challenge of generating detailed, short-term actions, Chen et al. [4] integrated value-learning policies [30] for low-level control. In another direction, Chen et al. [2] introduces causal noise scheduling with semi-autoregressive planning to improve performance in causality-sensitive tasks. More recently, research has explored leveraging increased computational budgets at inference time to enhance plan quality [33, 35]. Both Yoon et al. [33] and Zhang et al. [35] employ tree search methods within the denoising process. Yoon et al. [33] combines auto-regressive denoising with guidance scale-based action control for complex long-horizon planning, while Zhang et al. [35]

Table 2: **Maze results.** Success rates and planning times (± std) across PointMaze and AntMaze environments on medium, large giant sized map for *navigate* datasets. Best results are shown in strong positive colors, comparable ones in mild positives. For planning time, slowest results use strong negatives, and marginally slower ones use mild negatives.

| Env. | Method | Success Rate ↑ (%) | | | Planning Time ↓ (sec.) | | |
|------|--------|--------|-------|-------|--------|-------|-------|
| | | medium | large | giant | medium | large | giant |
| **PointMaze** | Diffuser | $58 \pm 6$ | $44 \pm 8$ | $0 \pm 0$ | $6.6 \pm 0.1$ | $6.5 \pm 0.1$ | $6.4 \pm 0.1$ |
| | Diffuser-Replan | $60 \pm 0$ | $40 \pm 0$ | $0 \pm 0$ | $40.6 \pm 2.7$ | $55.1 \pm 1.8$ | $137.8 \pm 0.4$ |
| | Diffusion Forcing | $65 \pm 16$ | $74 \pm 9$ | $50 \pm 10$ | $9.8 \pm 1.0$ | $12.5 \pm 0.5$ | $45.6 \pm 3.2$ |
| | MCTD | $100 \pm 0$ | $98 \pm 6$ | $100 \pm 0$ | $59.2 \pm 27.1$ | $174.6 \pm 27.2$ | $264.2 \pm 33.8$ |
| | P-MCTD (Ours) | $96 \pm 8$ | $100 \pm 0$ | $100 \pm 0$ | $7.4 \pm 1.1$ | $7.0 \pm 0.6$ | $9.8 \pm 1.3$ |
| | S-MCTD (Ours) | $100 \pm 0$ | $84 \pm 8$ | $100 \pm 0$ | $9.7 \pm 1.2$ | $226.1 \pm 114.4$ | $18.9 \pm 2.8$ |
| | **Fast-MCTD (Ours)** | $100 \pm 0$ | $80 \pm 0$ | $100 \pm 0$ | $3.6 \pm 0.6$ | $4.1 \pm 0.4$ | $2.4 \pm 0.1$ |
| **AntMaze** | Diffuser | $36 \pm 15$ | $14 \pm 16$ | $0 \pm 0$ | $6.2 \pm 0.1$ | $6.5 \pm 0.1$ | $6.5 \pm 0.1$ |
| | Diffuser-Replan | $40 \pm 18$ | $26 \pm 13$ | $0 \pm 0$ | $60.3 \pm 4.2$ | $73.7 \pm 1.9$ | $193.0 \pm 0.4$ |
| | Diffusion Forcing | $90 \pm 10$ | $57 \pm 6$ | $24 \pm 12$ | $14.0 \pm 3.8$ | $24.1 \pm 1.8$ | $67.0 \pm 3.4$ |
| | MCTD | $100 \pm 0$ | $98 \pm 6$ | $94 \pm 9$ | $68.0 \pm 3.8$ | $132.1 \pm 12.7$ | $214.5 \pm 11.0$ |
| | P-MCTD (Ours) | $96 \pm 8$ | $90 \pm 10$ | $89 \pm 16$ | $2.8 \pm 0.2$ | $3.1 \pm 0.1$ | $4.2 \pm 0.1$ |
| | S-MCTD (Ours) | $94 \pm 1$ | $68 \pm 13$ | $82 \pm 14$ | $12.5 \pm 0.7$ | $40.1 \pm 2.6$ | $40.3 \pm 2.9$ |
| | **Fast-MCTD (Ours)** | $98 \pm 6$ | $77 \pm 14$ | $90 \pm 16$ | $2.2 \pm 0.1$ | $2.2 \pm 0.1$ | $2.5 \pm 0.1$ |

develops a tree search method with learnable energy functions for value estimation, applying it to reasoning tasks such as Sudoku. However, as inference-time scaling in diffusion denoising has only recently been explored, optimizing efficiency in this context remains relatively understudied. Our work addresses this gap by drawing inspiration from established principles for improving efficiency, namely sparse planning and the acceleration of Monte Carlo Tree Search (MCTS).

Monte Carlo Tree Search (MCTS) [7] has a long history of success in decision-making domains, particularly when integrated with learned policies or value networks [24, 25]. It has also been applied to Large Language Models (LLMs) to enhance reasoning required task performances [31, 32, 34]. To address MCTS's inherent inefficiency in sequential search, various enhancements have been proposed, including parallel search methods [19], novel node selection policies [17], and visited-node penalization in parallel searches [25]. Our work adapts these established principles of parallelization and efficient exploration to the unique computational structure of diffusion-based planners.

# 6 Experiments

We evaluate Fast-MCTD using tasks from the Offline Goal-conditioned RL benchmark (OG-Bench) [22], aligning with the setup in MCTD [33]. These include tasks such as point and ant maze navigation with long horizons, robot arm multi-cube manipulation, and visually and partially observable mazes. In all tasks, we use the same guidance function following MCTD, $\mathcal{J}(\mathbf{x}) = \sum_{i=1}^{T} ||x_i - g||^2$ where $x_i$ is the $i$-th state and $g$ is the target goal. This design encourages trajectories that reach the goal as quickly as possible. For consistency and fair comparison, we use this distance-based guidance function for all baselines. The weight of $\mathcal{J}$, which serves as the guidance level, corresponds to the meta-action of MCTD and Fast-MCTD. Metrics reported are mean success rate (%) and planning time (sec), averaged over 50 runs (5 tasks × 10 seeds), including mean ± standard deviation. Detailed configurations and control settings are available in Appendix A.

**Baselines.** Besides MCTD, we include **Diffuser** [12], its replanning variant **Diffuser-Replan**, and **Diffusion Forcing** [2], which uses the causal noise schedule like MCTD but without explicit search.

## 6.1 Long-horizon planning in maze

We evaluate our methods' efficiency in extensive horizon scenarios using PointMaze and AntMaze tasks from OGBench [22], requiring up to 1000-step trajectories in *medium*, *large*, and *giant* mazes. Following [33], PointMaze employs a heuristic controller [12], and AntMaze uses a learned value-based policy [30]. Results in Table 2 highlight that Diffuser and Diffusion Forcing exhibit performance degradation as maze sizes increase, while MCTD maintains effectiveness but at significantly increased

Table 3: **Robot task results.** Success rates and planning times ($\pm$ std) across increasing task difficulty.

| Method | Success Rate ↑ (%) | | | Planning Time ↓ (sec.) | | |
|---|---|---|---|---|---|---|
| | Single | Double | Triple | Single | Double | Triple |
| Diffuser | $78 \pm 23$ | $12 \pm 10$ | $8 \pm 10$ | $6.3 \pm 0.1$ | $6.4 \pm 0.1$ | $6.5 \pm 0.1$ |
| Diffuser-Replan | $92 \pm 13$ | $12 \pm 13$ | $4 \pm 8$ | $21.4 \pm 2.4$ | $71.2 \pm 5.1$ | $130.9 \pm 11.6$ |
| Diffusion Forcing | $100 \pm 0$ | $18 \pm 11$ | $16 \pm 8$ | $2.9 \pm 0.2$ | $15.2 \pm 2.2$ | $15.9 \pm 1.2$ |
| MCTD-Replan | $100 \pm 0$ | $78 \pm 11$ | $40 \pm 21$ | $9.2 \pm 0.9$ | $38.8 \pm 4.7$ | $102.0 \pm 6.9$ |
| P-MCTD-Replan (Ours) | $100 \pm 0$ | $80 \pm 9$ | $50 \pm 21$ | $2.9 \pm 0.2$ | $6.5 \pm 2.3$ | $11.4 \pm 2.8$ |
| S-MCTD-Replan (Ours) | $100 \pm 0$ | $75 \pm 3$ | $42 \pm 11$ | $8.5 \pm 0.5$ | $35.7 \pm 18.2$ | $58.7 \pm 33.0$ |
| **Fast-MCTD-Replan (Ours)** | $100 \pm 0$ | $77 \pm 11$ | $50 \pm 16$ | $3.0 \pm 0.2$ | $5.9 \pm 1.7$ | $9.1 \pm 1.8$ |

Figure 3: **Planning visualization.** Sparse planning (top row) allows for more effective long-horizon trajectories than the dense (bottom row).

computational costs, illustrating the planning horizon dilemma clearly. For instance, MCTD exceeds 4 minutes to solve PointMaze-Giant.

In contrast, Fast-MCTD demonstrates approximately **80-110× speedups** relative to MCTD on giant maps with minimal performance loss. Surprisingly, Fast-MCTD is **even faster than Diffuser**, which employs straightforward denoising without search. Despite generally superior performance, Fast-MCTD encounters performance degradation on large maps due to sparse diffusion model limitations arising from insufficient start-position training data when subsampled every $H$ steps. Nevertheless, Sparse MCTD (S-MCTD) and Parallel MCTD (P-MCTD) achieve notable efficiency gains, in particular, P-MCTD reaches a 50× speedup on AntMaze-Giant.

## 6.2 Robot arm manipulation

We evaluate efficiency of our methods in compositional planning tasks involving multiple cube manipulations from OGBench [22]. Tasks require strategic manipulation sequences, such as stacking cubes in a specific order. High-level planning utilizes diffusion planners, and local actions are executed via value-learning policies [30], enhanced by object-wise guidance and replanning methods [33].

Fast-MCTD maintains or exceeds MCTD performance with significantly improved efficiency. Specifically, it shows better efficiency than Diffuser on single and double cube tasks. However, in triple cube tasks, Fast-MCTD is slower than Diffuser due to the increased task difficulty which demands more extensive search. Nevertheless, it remains approximately **10× faster than MCTD**, underscoring its significant computational advantages.

## 6.3 Visual planning

To assess efficiency in high-dimensional, partially observable contexts, we evaluate Fast-MCTD using visual maze environments where agents observe RGB image pairs depicting start and goal states [33]. Observations are encoded with a pretrained VAE [13], actions derived via inverse dynamics models in latent space, and approximate positional hints are provided by pretrained estimators.

As shown in Table 4, Fast-MCTD demonstrates a substantial efficiency improvement (**25-60× faster than MCTD**), even **outperforming it on larger mazes**. This is because the abstract trajectory planning through Sparse MCTD (S-MCTD) reduces rollout horizon complexity and improves credit assignment in latent transitions [18] as shown in Figure 3. Consequently, S-MCTD surpasses both MCTD and P-MCTD on larger maps. By integrating both sparse and parallel search, Fast-MCTD maximizes these efficiency gains. Replanning further enhances MCTD but at significant computational cost, whereas Fast-MCTD achieves superior performance with relatively minimal overhead, comparable to Diffuser-Replan.

Table 4: **Visual maze results.** Mean success rate (%) and planning time (seconds) on medium and large mazes.

| Method | Success Rate ↑ (%) | | Planning Time ↓ (sec.) | |
| --- | --- | --- | --- | --- |
| | Medium | Large | Medium | Large |
| Diffuser | $8 \pm 13$ | $0 \pm 0$ | $7.0 \pm 0.3$ | $6.7 \pm 0.3$ |
| Diffuser-Replan | $8 \pm 10$ | $0 \pm 0$ | $26.4 \pm 1.0$ | $26.0 \pm 0.4$ |
| Diffusion Forcing | $66 \pm 32$ | $8 \pm 12$ | $14.5 \pm 1.7$ | $17.1 \pm 0.7$ |
| MCTD | $76 \pm 20$ | $2 \pm 6$ | $98.8 \pm 36.2$ | $320.1 \pm 7.6$ |
| P-MCTD (Ours) | $86 \pm 16$ | $0 \pm 0$ | $6.8 \pm 0.4$ | $7.6 \pm 0.1$ |
| S-MCTD (Ours) | $82 \pm 14$ | $31 \pm 16$ | $40.1 \pm 32.0$ | $168.7 \pm 46.2$ |
| **Fast-MCTD (Ours)** | $80 \pm 18$ | $32 \pm 19$ | $4.0 \pm 0.2$ | $5.1 \pm 0.3$ |
| MCTD-Replan | $86 \pm 13$ | $31 \pm 10$ | $129.8 \pm 53.1$ | $419.7 \pm 49.1$ |
| **Fast-MCTD-Replan (Ours)** | $84 \pm 28$ | $38 \pm 30$ | $14.3 \pm 2.3$ | $26.7 \pm 3.7$ |

Table 5: Planning times for Redundancy-Aware vs. Unaware Selection in PointMaze.

| | Planning Time ↓ (sec.) | | |
| --- | --- | --- | --- |
| Redundancy | Medium | Large | Giant |
| Aware | $8.5 \pm 1.6$ | $7.4 \pm 1.3$ | $12.8 \pm 2.5$ |
| Unaware | $6.6 \pm 1.0$ | $7.8 \pm 0.6$ | $18.4 \pm 0.9$ |

Table 6: Planning time with different weight $w$ in PointMaze-Giant.

| Weight ($w$) | Planning Time ↓ (sec.) |
| --- | --- |
| 0.0 | $221.3 \pm 109.6$ |
| 0.1 | $14.0 \pm 2.5$ |
| 1.0 | $12.8 \pm 2.3$ |
| 5.0 | $12.5 \pm 2.1$ |

## 6.4 Ablation studies

**Redundancy-unaware selection.**    To assess the impact of Redundancy-Aware Selection (RAS) on planning efficiency, we ablate the constraint that prevents multiple rollouts from expanding the same node simultaneously. In this redundancy-unaware selection variant, rollouts still share visitation counts but can repeatedly expand the same node. Table 5 presents planning times of P-MCTD across different PointMaze sizes, measured under a generous computational budget with 100% success rates. By avoiding repeated expansion of the same node, RAS prevents excessive exploitation, which can dominate exploration when paths are harder to discover, especially in larger environments. In smaller environments, simple exploitation can still be effective, making the benefit of RAS less pronounced.

**Visitation count weight.**    We further investigate the impact of RAS during the selection phase by ablating the visitation count weight $w$ from Equation (5), which balances exploration and exploitation in parallel search. Table 6 presents the planning times in the PointMaze-Gaint for different $w$ values. In the standard UCT setting ($w = 0$), which lacks visitation count sharing, planning times exceed 220 seconds with high variance. In contrast, with RAS, planning times remain consistently low (12-14 seconds) across a wide range of $w$ values, suggesting that redundancy awareness is more critical than fine-tuning this hyperparameter.

**Parallelism degree and interval size.**    We also study two key hyperparameters: the parallelism degree (number of concurrent rollouts) and the subsampling interval size (steps skipped between coarsen states). Figure 4 shows the performance of Fast-MCTD in different configurations. Increasing the parallelism degree initially reduces planning time through efficient parallelization, but high degrees (above 200) degrade success rates due to delayed tree updates and selection redundancy. For interval size, larger interval sizes significantly boost efficiency but risk performance loss if the abstractions become too coarse for accurate low-level control [33].

## 7 Limitations and discussion

While our Fast Monte Carlo Tree Diffusion framework significantly enhances performance, it introduces certain trade-offs. Optimal parallelization for P-MCTD benefits from high-performance computing, yet substantial efficiency gains are achievable even with modest parallel capabilities on standard hardware. Our approach also introduces new hyperparameters, namely the parallelism degree and interval size. However, our ablation studies confirm the robustness of these parameters and

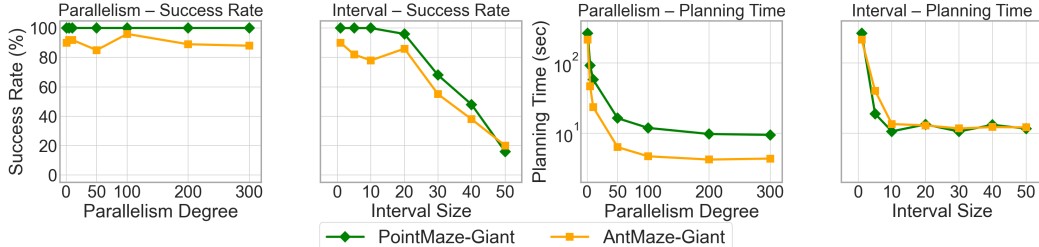

Figure 4: **Ablation studies for parallelism degree and interval size.** Success rates (%) and planning time (seconds) as the parallelism degree and interval size increase for maze giant tasks.

provide clear guidelines for their effective configuration. Furthermore, our sparse planning approach requires training a dedicated diffusion model on coarsened trajectories, which represents an additional upfront computational cost. The computational advantages of Fast-MCTD substantially outweigh the minor overhead of parameter tuning and additional training, enabling the practical deployment of diffusion-based planning in time-sensitive applications where it was previously infeasible.

# 8 Conclusion

We introduced Fast Monte Carlo Tree Diffusion (Fast-MCTD), a framework that resolves the critical efficiency bottleneck of Monte Carlo Tree Diffusion (MCTD), a state-of-the-art method for inference-time scalable planning. By synergizing parallelized tree search (P-MCTD) with sparse trajectory abstraction (S-MCTD), our method unlocks significant inference-time acceleration without compromising planning quality. Our experiments demonstrate that Fast-MCTD achieves up to a 100-fold speedup over prior work on challenging long-horizon tasks, such as maze navigation and robotic manipulation, while maintaining or even improving performance. These gains are driven by algorithmic innovations like search-aware parallel rollouts and a coarse-to-fine diffusion process over abstract plans. While Fast-MCTD alleviates key efficiency bottlenecks, future work could explore further improvements by integrating learned value function. Ultimately, our findings demonstrate that test-time scalability and structured reasoning are not mutually exclusive, opening new avenues for developing fast, deliberative agents in high-dimensional domains.

## Acknowledgments and Disclosure of Funding

We are grateful to Doojin Baek and Junyeong Park for their insightful discussions and valuable assistance with this project. This research was supported by GRDC (Global Research Development Center) Cooperative Hub Program (RS-2024-00436165) through the National Research Foundation of Korea (NRF) funded by the Ministry of Science and ICT (MSIT) and Basic Research Laboratory Program (No. RS-2024-00414822) through the National Research Foundation of Korea (NRF) grant funded by the Korea government (MSIT) and Institute of Information & communications Technology Planning & Evaluation (IITP) grant funded by the Korea government (MSIT) (No. RS-2024-00509279, Global AI Frontier Lab). YB acknowledges the support from CIFAR, NSERC, the Future of Life Institute.

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

Figure 5: **Task Illustrations**: (a) long-horizon maze navigation, (b) multi-object robotic manipulation requiring compositional planning, and (c) visual maze planning from raw RGB observations.

## A  Experiment details

### A.1  Computation resources

The experiments were conducted on a server equipped with 8 NVIDIA RTX 4090 GPUs, 512 GB of system memory, and a 96-thread CPU. Model training required approximately 3–6 hours per model, while inference for each evaluation took up to 5 minutes.

### A.2  Environment details

Following prior work [33], we evaluate our methods (P-MCTD, S-MCTD, and Fast-MCTD) on challenging scenarios from the OGBench [22] goal-conditioned benchmark. These include long-horizon maze navigation, robot arm manipulation, and visual maze navigation, as illustrated in Figure 5. All evaluation tasks were unseen during training, requiring the models to generalize their planning capabilities to novel scenarios at inference time.

#### A.2.1  Long-horizon maze

To assess performance over long planning horizons, we test our methods on point-mass and ant robot navigation in three increasingly complex environments: medium, large, and giant mazes. The models were trained on the *navigate dataset*, which contains long trajectories but lacks specific task information. At inference, the models must leverage their learned dynamics to solve these unseen maze configurations, thereby testing their generalization for complex, long-horizon problems.

#### A.2.2  Robot arm manipulation

The robot manipulation tasks demand compositional planning to move multiple objects in a specific sequence. For instance, when tasked with stacking objects in a predetermined order, the model must generate a coherent plan that adheres to these sequential constraints. As shown in prior work [33], holistic diffusion-based planners often struggle with such compositional tasks without explicit search or reward guidance. We evaluated Fast-MCTD in this domain, focusing on improving efficiency within a replanning framework while preserving MCTD's strong compositional planning performance.

#### A.2.3  Visual maze

Following Yoon et al. [33], we evaluate performance in high-dimensional, partially observable settings using the Visual Antmaze task from OGBench [22]. In this task, the agent navigates from an initial observation to a goal, where both are represented as $64 \times 64$ RGB images. These images are encoded into an 8-dimensional latent space using a Variational Autoencoder (VAE) [13], providing a compact representation for efficient planning.

Table 7: Hyperparameter configurations for the Diffuser.

| Hyperparameter | Value |
|---|---|
| *Training configuration* | |
| Learning rate | $2 \times 10^{-4}$ |
| EMA decay | 0.995 |
| Precision (training/inference) | 32-bit (FP32) |
| Batch size | 32 |
| Max training steps | 20,000 |
| *Diffusion & guidance* | |
| Beta schedule | Cosine |
| Diffusion objective | $x_0$-prediction |
| Guidance scale | 0.1 |
| *Planning configuration* | |
| Planning horizon | Task-dependent (see Sec A.7) |
| Open loop horizon | 50 (for replanning) |
| *Model architecture (U-Net)* | |
| Depth | 4 |
| Kernel size | 5 |
| Channel dimensions | (32, 128, 256) |

## A.3 Baselines

We compared Fast-MCTD against several diffusion-based planning approaches:

- **Diffuser** [12]: We implemented the standard Diffuser with classifier-guided generation [8] as a baseline, ensuring comparable guidance mechanisms across all methods.

- **Diffuser-Replan**: To strengthen the Diffuser baseline, we incorporated periodic replanning (details are in Appendix A.4). This variant regenerates the entire plan at fixed intervals, allowing it to adapt to intermediate state observations.

- **Diffusion Forcing** [2]: This approach utilizes causal noise scheduling with tokenized trajectories but omits an explicit search mechanism. Its default implementation includes replanning, making it a strong comparative baseline.

- **MCTD** [33]: As our primary baseline, MCTD represents the state-of-the-art in inference-time scaling for diffusion planning. It integrates causal noise scheduling with a tree-based search. While effective, it exhibits significant computational inefficiencies in complex planning scenarios.

- **MCTD-Replan**: This variant applies the replanning strategy (Appendix A.4) to MCTD, allowing it to adapt to intermediate state observations while maintaining high-quality planning. However, this process incurs significant computational overhead compared to other baselines, serving as a powerful but costly baseline.

## A.4 Replanning strategy

The replanning strategy is applied to our replanning-based baselines: Diffuser-Replan, Diffusion Forcing, and MCTD-Replan. In this variant, the agent generates a new full-horizon plan (e.g., 500 steps in length) at each interval but executes only the first 50 steps before replanning from the updated state. This process is repeated until the task is completed.

## A.5 Model hyperparameters

We maintain consistency with the hyperparameter configurations from prior work [33] and describe the detailed hyperparameters here to enhance reproducibility. We highlight the key parameters specific to our methods; complete implementation details will be made available.

Table 8: Hyperparameters for the value-learning policy.

| Hyperparameter | Value |
|---|---|
| *Optimizer settings* | |
| Learning rate | $3 \times 10^{-4}$ |
| Gradient clipping norm | 7.0 |
| Learning eta | 1.0 |
| *Training strategy* | |
| Training epochs | 2,000 |
| Target steps | 10 |
| Data sampling randomness ($p$) | 0.2 |
| Reward tuning | cql_antmaze |
| *Q-learning details* | |
| Max Q backup | False |
| Top-k | 1 |

Table 9: Hyperparameters for the Diffusion Forcing.

| Hyperparameter | Value |
|---|---|
| *Training & Optimizer Settings* | |
| Learning Rate | $5 \times 10^{-4}$ |
| Weight Decay | $1 \times 10^{-4}$ |
| Warmup Steps | 10,000 |
| Batch Size | 1024 |
| Max Training Steps | 200,005 |
| Training Precision | 16-bit Mixed |
| Inference Precision | 32-bit (FP32) |
| *Model Architecture (Transformer)* | |
| Hidden Size | 128 |
| Number of Layers | 12 |
| Attention Heads | 4 |
| Feedforward Dimension | 512 |
| Frame Stack Size | 10 |
| Causal Masking | Not Used |
| *Diffusion & Sampling Settings* | |
| Beta Schedule | Linear |
| Objective | $x_0$-prediction |
| Scheduling Matrix | Pyramid |
| DDIM Sampling $\eta$ | 0.0 |
| *Planning & Guidance Settings* | |
| Planning Horizon | Task-dependent (see Sec A.7) |
| Open-loop Horizon | 50 |
| Guidance Scale | 3.0 (medium mazes), 2.0 (others) |
| Stabilization Level | 10 |

## A.6 Diffusion model training protocol

We follow the training protocol established by prior diffusion planners [2, 12]. Concretely, the model is trained on offline trajectory datasets from OGBench [22]. The training objective is a standard denoising loss (e.g., L2 loss), where the model learns to predict the original trajectory $\mathbf{x}_0$ from a noised version. We employ a Transformer-based architecture. All hyperparameters, including learning rates and batch sizes, are fully reported in Appendix A.5.

Table 10: Hyperparameters for MCTD and its variants.

| Hyperparameter | Value |
|---|---|
| *Training & Optimizer Settings* | |
| Learning Rate | $5 \times 10^{-4}$ |
| Weight Decay | $1 \times 10^{-4}$ |
| Warmup Steps | 10,000 |
| Batch Size | 1024 |
| Max Training Steps | 200,005 |
| Training Precision | 16-bit Mixed |
| Inference Precision | 32-bit (FP32) |
| *Model Architecture (Transformer)* | |
| Hidden Size | 128 |
| Number of Layers | 12 |
| Attention Heads | 4 |
| Feedforward Dimension | 512 |
| Frame Stack Size | 10 |
| Causal Masking | Not Used |
| *Diffusion & Sampling Settings* | |
| Beta Schedule | Linear |
| Objective | $x_0$-prediction |
| Scheduling Matrix | Pyramid |
| Partial Denoising Steps | 20 |
| Jumpy Denoising Interval | 10 |
| DDIM Sampling $\eta$ | 0.0 |
| *Planning & Search Settings* | |
| Max Search Iterations | 500 |
| Stabilization Level | 10 |
| Planning Horizon | Task-dependent (see Sec A.7) |
| Open-loop Horizon | 50 (for replanning), otherwise full horizon |
| Guidance Set | Task-dependent (see Sec A.7) |
| ***Method-Specific Settings*** | |
| **Parallelism Degree** | **200** (P-MCTD, Fast-MCTD), **1** (otherwise) |
| **Parallel Search Weight** ($w$) | **1.0** |
| **Subsampling Interval** ($H$) | **5** (S-MCTD, Fast-MCTD), **1** (otherwise) |

## A.7 Evaluation details

Our experimental setup, including all hyperparameters and environmental settings, is consistent with prior work [33]. The specific configurations for each task are detailed below.

### A.7.1 Maze navigation with point-mass and ant robots

**Planning parameters.** For the point-mass maze tasks, we use a guidance scale set of $\{0, 0.1, 0.5, 1, 2\}$, while for the antmaze evaluations, the set is $\{0, 1, 2, 3, 4, 5\}$. The planning horizon is set to 500 for medium and large mazes and is extended to 1000 for giant mazes.

**Low-level controller.** For the point-mass maze tasks, we employ the heuristic controller from Janner et al. [12]. For the antmaze tasks, we utilize the diffusion-based value-learning policy from Chen et al. [4], which functions as a low-level controller to navigate the agent toward a given subgoal. A new subgoal is assigned every 10 steps, contingent upon the successful arrival at the previous one.

**Reward function for tree search-based planners.** For the maze navigation tasks, we adopt the reward function from Yoon et al. [33]. This function assigns a reward of zero to physically implausible trajectories, such as those with excessively large changes between consecutive states. For a valid trajectory that reaches the goal at timestep $t$, the reward is calculated as $r = (H`t)/H$, where H

Table 11: **Leaf parallelization ablation study results.** Success rates and planning times ($\pm$ std) across PointMaze environment on medium, large giant sized maps.

| Env. | Method | Success Rate ↑ (%) | | | Planning Time ↓ (sec.) | | |
|---|---|---|---|---|---|---|---|
| | | medium | large | giant | medium | large | giant |
| **PointMaze** | No Leaf Par. (P-MCTD) | $100 \pm 0$ | $100 \pm 0$ | $100 \pm 0$ | $8.5 \pm 1.6$ | $7.4 \pm 1.3$ | $12.8 \pm 2.5$ |
| | Leaf Par. | $98 \pm 6$ | $100 \pm 0$ | $100 \pm 0$ | $7.4 \pm 0.9$ | $6.7 \pm 0.6$ | $17.0 \pm 2.8$ |

is the maximum horizon length. This reward structure incentivizes the discovery of shorter, more efficient paths.

### A.7.2 Robot arm cube manipulation

**Planning parameters and low-level controller.** For each object, we utilize a set of guidance scales $\{1, 2, 4\}$. As MCTD employs object-wise guidance [33], the total number of guidance combinations scales with the number of objects. The planning horizon is set to $H = 200$ for single-cube tasks and $H = 500$ for all multi-cube manipulation tasks. We employ the same value-learning policy as in the antmaze tasks, following Wang et al. [30].

**Redundant plans.** Consistent with Yoon et al. [33], we implement redundant plans to resolve inconsistencies arising from object-wise planning. These plans are inserted between object-wise plan segments and execute pre-defined actions, such as opening the gripper to release an object or returning to a default position. This prevents infeasible actions, for instance, attempting to grasp a new object while another is already held or moving to a distant, unrelated location.

**Reward function for tree search-based planners.** We adopt the reward function from Yoon et al. [33] for the robot arm manipulation tasks. This function assigns a reward of zero to physically implausible outcomes, including: (1) moving multiple cubes simultaneously; (2) leaving a cube suspended in mid-air; (3) causing collisions between cubes; or (4) grasping a cube that is obstructed by another. For trajectories that successfully complete the task at timestep $t$, a positive reward of $r = (H`t)/H$ is assigned, where $H$ is the maximum horizon length.

### A.7.3 Visual point-mass maze

**Framework for visual environment.** For the visual maze tasks, we introduce a framework consisting of three pre-trained components. First, a Variational Autoencoder (VAE) [13] is pre-trained to encode visual observations into a compact 8-dimensional latent representation, $z$. This representation serves as the input for our planning and control models. Second, an MLP-based inverse dynamics model, $f_{\text{inv}}$, is pre-trained to predict the action $\hat{a}_t$ required for a transition. To infer velocity from static observations, the model is conditioned on three consecutive latent states: $\hat{a}_t = f_{\text{inv}}(z_{t-1}, z_t, z_{t+1})$. This model functions as the low-level controller. Third, we pre-train an MLP-based position estimator to predict the agent's coordinates from the latent state $z$. This estimator provides a positional signal for planning guidance without compromising the task's partial observability.

**Planning parameters.** We use a set of guidance scales $\{0, 0.1, 0.5, 1, 2\}$. Guidance is applied in the state space using the coordinates inferred by the pre-trained position estimator. This approach allows us to employ the same guidance function used in our other tasks (Section 6). A fixed planning horizon of $H = 500$ is used for both the medium and large maze environments.

## B Additional experimental results

### B.1 Leaf parallelization ablations

We conduct an ablation study to compare our method against **leaf parallelization**, a common MCTS strategy that expands multiple children from a single node in parallel [19]. The motivation for this approach is to reduce selection overhead by performing multiple expansions per selection operation.

As shown in Table 11, this strategy exhibits inconsistent performance across environment scales. While it offers marginal speedups in the **medium** (8.5s vs. 7.4s) and **large** (7.4s vs. 6.7s) mazes, it

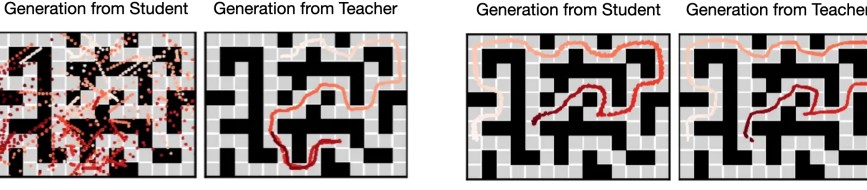

| Generation from Student | Generation from Teacher | | Generation from Student | Generation from Teacher |

(a) Varying noise levels per subplan     (b) Uniform noise levels per subplan

Figure 6: **Distillation challenges in MCTD.** Trajectory quality from distilled diffusion models varies significantly based on the noise level distribution during training. (a) Using variable noise levels (required for causal scheduling) leads to poor trajectory quality. (b) Using uniform noise levels produces cleaner trajectories but is incompatible with MCTD's causal scheduling requirements.

significantly increases planning time in the **giant** maze (12.8s vs. 17.0s), degrading performance in the most computationally demanding setting.

We attribute this degradation to a disruption of the exploration-exploitation balance managed by our search-aware selection mechanism (Equation 5). By forcing the expansion of multiple children from the same node, leaf parallelization concentrates computational resources on potentially suboptimal branches. This effect is particularly detrimental in larger search spaces where strategic exploration is crucial. These findings validate our design choice to parallelize across disparate tree branches rather than within a single node's children, especially for complex, long-horizon planning tasks.

### B.2 Distillation of diffusion planner ablations

We investigated **diffusion model distillation**, a prominent technique for accelerating the denoising process [23], as a potential optimization for MCTD. We trained a student model to predict the output of two teacher denoising steps, following the progressive distillation method.

However, this approach produced low-quality trajectories (Figure 6a). We posit that this failure stems from a fundamental incompatibility: standard distillation methods assume a **uniform noise level** across the entire data sample, whereas MCTD's causal scheduling applies **heterogeneous, per-subplan noise levels**. This transforms the distillation task into a more complex learning problem, as the student must learn a mapping that is conditioned on a non-uniform noise schedule.

Our findings suggest that successfully distilling MCTD is a non-trivial task that requires novel distillation techniques capable of handling such heterogeneous noise distributions. We leave the development of these methods to future work.

## C Further discussion

### C.1 On the choice of fixed-interval subsampling

S-MCTD employs a fixed-interval subsampling strategy for temporal abstraction. This approach contrasts with the semantic or hierarchical subgoal definitions typical of the classical options framework [29], yet aligns with recent trends in diffusion-based planners, such as Hierarchical Diffusers [5, 9] and PlanDQ [4]. This design choice provides a simple yet general mechanism for abstracting trajectories.

Our empirical results corroborate the efficacy of this fixed-interval scheme. Notably, prior work [5] has shown that fixed-interval subsampling can outperform adaptive subgoal selection methods like HDMI [15]. These findings suggest that a simple, uniform subsampling strategy can be more effective than complex, learned subgoal discovery techniques in certain domains.

### C.2 Limitations and trade-offs of sparse planning

While sparse planning enhances the efficiency of Fast-MCTD, the level of abstraction, controlled by the hyperparameter $H$, introduces a trade-off with performance. We empirically identified two primary failure modes:

1. **Excessively coarse abstraction:** As shown in Figure 4, the success rate degrades significantly when the abstraction interval $H$ exceeds 20. This trend, observed across both PointMaze-Giant and AntMaze-Giant, highlights the limitations of sparse guidance in scenarios requiring fine-grained control.

2. **Tasks requiring high precision:** As reported in Table 2, S-MCTD achieves a 100% success rate in PointMaze-Medium and Giant but drops sharply to 8% in PointMaze-Large, despite using an identical planning architecture. We hypothesize this discrepancy is due to the task's geometric constraints, such as narrow corridors, which demand high-precision maneuvering. The coarse temporal abstraction may cause the planner to overlook critical intermediate states required to navigate these tight spaces successfully.

## C.3 Generalization

A key feature of Fast-MCTD is its ability to perform high-quality planning efficiently at inference time. This design enables zero-shot adaptation to novel goals and initial states without requiring any retraining. However, consistent with other models trained on a fixed data distribution [2, 12, 33], its performance may degrade when deployed in environments with significant dynamics shifts or out-of-distribution states.

## C.4 Future work

A promising avenue for future research is the integration of learned value functions or policy priors to guide the search process. Such guidance could significantly enhance planning efficiency by pruning unpromising branches of the search tree, thereby reducing redundant rollouts. However, a key challenge lies in integrating these components without compromising the simplicity and parallel scalability of Fast-MCTD's sparse planning framework.

Another compelling direction is extending Fast-MCTD to multi-agent planning domains. While the core framework is theoretically applicable, a naive extension would face challenges from the combinatorial explosion of the joint action and state spaces. Here, learned value functions or factored representations could be crucial for tractably managing the search complexity.

Finally, exploring more flexible temporal abstractions is a promising direction. While S-MCTD uses a fixed-interval subsampling for its simplicity and parallelization benefits, adaptive interval selection could dynamically enhance efficiency and accuracy. This might mitigate over-abstraction issues in tasks requiring high precision. However, this must be balanced against added complexity and findings that simple fixed-interval schemes can outperform adaptive methods in some cases. Investigating this trade-off remains a valuable area for future research.

# D Algorithms

---

**Algorithm 2** Fast Monte Carlo Tree Diffusion

---

1: **procedure** FAST-MCTD($root, iterations, parallelism\_degree, jump\_interval$)
2:     **for** $i = 1$ to $iterations$ **by** $parallelism\_degree$ **do**
3:         $nodes\_to\_expand \leftarrow [\,]$
4:         $temp\_visit\_counts \leftarrow \{\}$              ▷ Track parallel selections
5:         **for** $j = 1$ to $parallelism\_degree$ **do**         ▷ Search-aware selection
6:             $node \leftarrow root$
7:             **while** ISFULLYEXPANDED($node$) **and not** ISLEAF($node$) **do**
8:                 $node \leftarrow$ SEARCHAWAREUCT($node, temp\_visit\_counts$)
9:                 $temp\_visit\_counts[node] \leftarrow temp\_visit\_counts.get(node, 0) + 1$
10:             **end while**
11:             APPEND($nodes\_to\_expand, node$)
12:         **end for**
13:         $expansions \leftarrow [\,]$
14:         **for** $node \in nodes\_to\_expand$ **in parallel do**     ▷ Parallel expansion phase
15:             **if** ISEXPANDABLE($node$) **then**
16:                 $g_s \leftarrow$ SELECTMETAACTION($node$)
17:                 APPEND($expansions, (node, g_s)$)
18:             **end if**
19:         **end for**
20:         $new\_children \leftarrow$ BATCHDENOISESUBPLANS($expansions, jump\_interval$)
21:         **for** $(node, child) \in new\_children$ **do**
22:             ADDCHILD($node, child$)
23:         **end for**
24:         $simulations \leftarrow [\,]$
25:         **for** $(node, child) \in new\_children$ **in parallel do**   ▷ Parallel simulation phase
26:             $partial \leftarrow$ GETPARTIALTRAJECTORY($child$)
27:             APPEND($simulations, (node, child, partial)$)
28:         **end for**
29:         $rewards \leftarrow$ BATCHFASTSPARSEDENOISING($simulations, jump\_interval$)
30:         **for** $(node, child, reward) \in rewards$ **do**        ▷ Delayed tree update
31:             $current \leftarrow child$
32:             **while** $current \neq null$ **do**
33:                 $current.visitCount \leftarrow current.visitCount + 1$
34:                 $current.value \leftarrow$ MAX($current.value, reward$)
35:                 UPDATEMETAACTIONSCHEDULE($current, reward$)
36:                 $current \leftarrow current.parent$
37:             **end while**
38:         **end for**
39:     **end for**
40:     **return** BESTCHILD($root$)
41: **end procedure**

---

