# OpenReview forum: "Fast Monte Carlo Tree Diffusion: 100× Speedup via Parallel and Sparse Planning"
_NeurIPS.cc/2025/Conference — NeurIPS 2025 spotlight_

### Official Review · Reviewer_EsS3 · 2025-06-15

**Clarity:** 4
**Significance:** 3
**Originality:** 3
**Rating:** 6
**Confidence:** 4

**Summary:**

Fast-MCTD combines two enhancements to the original MCTD: Parallel MCTD (P-MCTD), which introduces parallel rollouts with delayed tree updates and redundancy-aware selection to minimize duplicated computation; and Sparse MCTD (S-MCTD), which reduces planning complexity by operating on coarsened trajectory segments. The core innovation lies in integrating these methods to preserve MCTD’s planning strength while enabling high-throughput and low-latency execution, leveraging GPU acceleration and hierarchical abstraction.

The method is evaluated on OGBench tasks, including PointMaze, AntMaze, robot arm manipulation, and visual planning. Comparisons are made against Diffuser, Diffusion Forcing, and MCTD using metrics like success rate and planning time. Ablation studies isolate the impact of key design choices.

**Questions:**

1. How does Fast-MCTD scale with increasing action/state space complexity beyond current benchmarks? Would it remain computationally tractable in domains like multi-agent planning?
2. What is the role of learned value functions or priors in further enhancing the efficiency of search within Fast-MCTD? Can these be integrated without negating the gains from sparse and parallel planning?
3. Could adaptive interval selection in S-MCTD improve both efficiency and accuracy dynamically? This would potentially mitigate the issue of over-abstraction in long-horizon tasks.

**Ethical Concerns:**

["NO or VERY MINOR ethics concerns only"]

**Final Justification:**

This paper deserves strong acceptance due to its rigorous method design and extensive evaluations. This could benefit many works using MCTS in goal-conditioned RL settings.

**Limitations:**

Yes

**Quality:**

3

**Strengths And Weaknesses:**

**Strengths**
- **Algorithmic Design:** The paper proposes Fast-MCTD, integrating two novel variants—Parallel MCTD and Sparse MCTD—which address key inefficiencies in Monte Carlo Tree Diffusion (MCTD).
- **Significant Efficiency Gains:** Fast-MCTD achieves up to 100× speedup over MCTD and even outperforms purely denoising-based approaches like Diffuser in some inference scenarios.
- **Thorough Empirical Validation:** The paper evaluates its approach across diverse domains (maze navigation, robotic manipulation, visual planning), and performs extensive ablation studies on key components (e.g., Redundancy-Aware Selection, parallelism degree). It benchmarks against relevant state-of-the-art models such as Diffuser, Diffusion Forcing, and MCTD, ensuring a meaningful performance context.
- **Clear Theoretical Motivation:** The authors discuss the “Planning Horizon Dilemma” and analyze computational complexity, helping justify the need for both parallelization and trajectory coarsening.

**Weaknesses**
- **Limited Discussion on Generalization:** While efficiency and performance gains are clear in the tested domains, the paper could elaborate more on how well Fast-MCTD generalizes to unseen environments or out-of-distribution tasks.
- **Sparse Planning Trade-offs:** The use of trajectory coarsening (S-MCTD) introduces approximation. While the authors do provide some mitigation strategies and ablations, a deeper analysis of where abstraction breaks down would be valuable.

---

> ### Author Rebuttal · Authors · 2025-07-31
>
> We thank reviewer EsS3 for insightful comments and constructive questions regarding potential directions for further improvement. Below, we address these points and outline how we will incorporate them into the revised manuscript.
>
> > Limited Discussion on Generalization: While efficiency and performance gains are clear in the tested domains, the paper could elaborate more on how well Fast-MCTD generalizes to unseen environments or out-of-distribution tasks.
>
> We agree that explicitly discussing generalization is important.
>
> Although our current experiments follow the standard evaluation protocols in this line of research, we also agree that these protocols should be improved to better assess generalization to unseen environments.
>
> Fast-MCTD is designed to efficiently perform tree-search-based planning entirely at inference time. **This enables adaptation to novel goals and initial states without retraining.** However, **like other diffusion-based planners**, Fast-MCTD may face **limitations when transferred to environments** with fundamentally different dynamics or state distributions. We will clarify these points in the revised manuscript.
>
> > Sparse Planning Trade-offs: The use of trajectory coarsening (S-MCTD) introduces approximation. While the authors do provide some mitigation strategies and ablations, a deeper analysis of where abstraction breaks down would be valuable.
>
> We thank the reviewer for this important observation. We empirically identified two primary conditions under which abstraction begins to fail:
>
> 1. **Overly large waypoint intervals**: As shown in Figure 4 , success rate degrades significantly when the abstraction interval increases beyond 20. This trend holds across both PointMaze-Giant and AntMaze-Giant, reflecting the limits of sparse guidance when dense, fine-grained control is needed.
> 2. **Precision-sensitive tasks**: As shown in the main results table, S-MCTD achieves perfect success (100%) in PointMaze-medium and giant, but drops sharply to 84% in PointMaze-large, despite using the same planning architecture. We attribute this to the precision demands of PointMaze-large, where narrow corridors and sharp turns make the agent more sensitive to omitted intermediate states. The coarse abstraction in S-MCTD likely causes the planner to miss key maneuvering steps.
>
> In our revision, we will provide a deeper analysis illustrating these boundaries more explicitly.
>
> > What is the role of learned value functions or priors in further enhancing the efficiency of search within Fast-MCTD? Can these be integrated without negating the gains from sparse and parallel planning?
>
> **We view this as a highly interesting avenue for future work.** Incorporating learned value functions or priors is indeed a promising direction for further improving the planning efficiency of Fast-MCTD. Value functions or learned priors can help bias the search toward more promising regions, reducing redundant rollouts and potentially increasing both sample efficiency and scalability. However, integrating these techniques with sparse and parallel planning in diffusion-based search is nontrivial, as it requires careful coordination to avoid undermining the scalability and simplicity that Fast-MCTD offers. We will mention it as a potential extension in the revised manuscript.
>
> > Would it remain computationally tractable in domains like multi-agent planning?
>
> We acknowledge that **extending Fast-MCTD to multi-agent planning is an open challenge.** While the framework is applicable to multi-agent domains “in principle”, computational efficiency may deteriorate due to the combinatorial growth of joint action and state spaces. Incorporating learned value functions might help mitigate these challenges, and we will discuss such extensions as promising future directions.
>
> > How does Fast-MCTD scale with increasing action/state space complexity beyond current benchmarks?
>
> We acknowledge that **applying Fast-MCTD to significantly larger action/state spaces remains to be thoroughly validated.** However, our current experiments already involve high-dimensional settings, such as visual mazes and giant PointMaze, and we expect Fast-MCTD to scale reasonably well. That said, further scaling may introduce new challenges, which we consider an important direction for future validation.
>
> > Could adaptive interval selection in S-MCTD improve both efficiency and accuracy dynamically? This would potentially mitigate the issue of over-abstraction in long-horizon tasks.
>
> We thank the reviewer for this valuable suggestion. We agree that adaptive interval selection in S-MCTD could potentially improve both efficiency and accuracy by dynamically adjusting the abstraction level.
>
> However, prior works [1, 2] empirically showed that fixed-interval abstractions can outperform adaptive subgoal selection methods like HDMI [3]. This suggests that while adaptive methods offer flexibility, they may also introduce additional overhead or instability, especially in high-parallelism settings like Fast-MCTD.
>
> Nonetheless, we believe that developing smart adaptive abstraction methods remains a promising direction, which can further enhance the performance of Fast-MCTD. We will include this as an exciting future direction in the revised manuscript.
>
> [1] Simple Hierarchical Planning with Diffusion., Chen et al., ICLR’24
>
> [2] DiffuserLite: Towards Real-time Diffusion Planning, Dong et al., NeurIPS’24
>
> [3] Hierarchical diffusion for offline decision making, Li et al., ICML’23

---

> > ### Comment · Reviewer_EsS3 · 2025-08-02
> >
> > I thank the authors for addressing my remaining concerns. Hence, I raised my score.

---

> > > ### Author Response · Authors · 2025-08-05
> > >
> > > We extend our deepest gratitude to the reviewer for their constructive feedback and positive reassessment of our work. Their guidance has significantly strengthened both our discussion and experimental analysis.

---

### Official Review · Reviewer_pbF8 · 2025-07-02

**Clarity:** 4
**Significance:** 3
**Originality:** 2
**Rating:** 5
**Confidence:** 4

**Summary:**

Recent work (MCTD) has used pretrained diffusion models to enable long-horizon planning using partially denoised plans along with Monte-Carlo tree search (MCTS). This work proposes to speed up MCTD using both parallel rollouts and a sparse trajectory-coarsening approach. Experiments with goal-conditioned RL tasks show that method achieves state-of-the-art results compared to other diffusion-based planning approaches. Both the parallel rollouts and the sparse trajectory coarsening approach maintains the good performance of MCTD while improving timing by up to two orders of magnitude.

**Questions:**

It seems like the experimental setup has been established in previous works such as MCTD and others. However, for the work to be self-contained, I would suggest explicitly defining $\mathcal{J}_\phi$ from Eq. 2 for each of the tasks. Is the "guidance set" weights of $\mathcal{J}_\phi$ that correspond to the actions of the MCTS? What are the number of timesteps in the pretraining trajectories and how do they compare to the number of timesteps in the found plans? More explanation of the "replanning" model variants would be helpful. Overall, I think the paper would be improved if details such as these were made more explicit in the main text.

**Ethical Concerns:**

["NO or VERY MINOR ethics concerns only"]

**Final Justification:**

After discussion with the authors, I remain convinced this is a strong paper. The original manuscript lacked clarity in some details, but the proposed clarification should fix this issue.

**Limitations:**

Overall, yes, but see the last paragraph of "Strengths and Weaknesses."

**Quality:**

4

**Strengths And Weaknesses:**

Overall, the paper is written very clearly and the main contributions and experimental results are easy to read and understand. The two proposed modifications to MCTD are well-motivated and the ablation experiments are well done and give the reader a good sense of how the additional hyperparameters affect planning capacity (Tables 5, 6, Figure 4).

The performance of the model is very good and the problem of planning with pretrained diffusion models is an important research direction, especially given the recent uptick in interest in the flexible use of pretrained models with inference-time compute.

The paper is only moderately original, trajectory coarsening is not very novel in this context and parallel MCTS is a straightforward design improvement. I view this as a minor weakness of the paper.

Another weakness of the proposed approach is that the trajectory coarsening approach requires training a new diffusion model on coarsened trajectories, so the overall method is not strictly inference-time. This should be added to the discussion of limitations.

---

> ### Author Rebuttal · Authors · 2025-07-31
>
> > The paper is only moderately original, *trajectory coarsening is not very novel* in this context *and parallel MCTS is a straightforward design improvement. I view this as a minor weakness of the paper.*
>
> We thank reviewer pbF8 for the thoughtful feedback.
>
> The main contribution of the paper is to propose a practical solution for improving the speed of MCTD, rather than introducing a conceptually novel approach. We acknowledge that, methodologically, the individual components may appear somewhat incremental. Nevertheless, we believe that their joint adaptation and integration specifically within the diffusion-based framework present a novel combination that effectively addresses key efficiency issues in MCTD.
>
> We will revise the manuscript to better highlight this distinction and the unique innovations introduced.
>
> > Trajectory coarsening approach requires training a new diffusion model on ccoarsened trajectories, so the overall method is not strictly inference-time. This should be added to the discussion of limitations.
>
> Thank you for the helpful comment. We agree that Fast-MCTD requires training a separate diffusion model on coarsened trajectories. While the planning process is entirely inference-time, the abstraction does require an additional training phase. We will clarify this point in the Limitations section.
>
> > It seems like the experimental setup has been established in previous works such as MCTD and others. However, for the work to be self-contained*,* I would suggest explicitly defining $\mathcal{J}$ that correspond to the actions of the MCTS from Eq.2 for each of the tasks .
>
> We thank the reviewer for highlighting the importance of clearly defining $\mathcal{J}$.
>
> In all tasks, we use the same guidance function following MCTD.
>
> $
> \mathcal{J}(\mathbf{x}) =\sum_{i=1}^T ||x_i - g||^2
> $
>
> where $x_i$ is the $i$-th state and $g$ is the target goal. This design encourages trajectories that reach the goal as quickly as possible. For consistency and fair comparison, we use this distance-based guidance function for all baselines and all task domains.
>
> We will make this definition explicit in the main text to improve clarity and self-containedness.
>
> > Is the "guidance set" weights of $*\mathcal{J}$* that correspond to the actions of the MCTS?
>
> Yes, in our framework, the “guidance set” refers to a set of discrete scalar weights applied to $\mathcal{J}$ during each tree expansion. Each weight functions as a meta-action, directly influencing the degree of goal-directedness for the generated trajectory segment at that node.
>
> In the MCTS process, selecting a specific weight from the guidance set at each branching point is analogous to choosing an action; this mechanism enables the planner to balance exploration and exploitation by adjusting the guidance strength.
>
> The guidance sets used in our experiments are as follows:
>
> - **PointMaze (medium/large):** $[0, 0.1, 0.5, 1, 2]$
> - **PointMaze (giant):** $[0.5, 1, 2, 3, 4]$
> - **AntMaze:** $[0, 1, 2, 3, 4, 5]$
>
> Each value is treated as a meta-action, and the MCTS expands the tree across these discrete choices.
>
> > What are the number of timesteps in the pretraining trajectories and how do they compare to the number of timesteps in the found plans? More explanation of the "replanning" model variants would be helpful.
>
> The pretrained diffusion model is trained on trajectory segments of 500 timesteps for medium and large mazes, and 1000 timesteps for giant mazes. At inference time, the planner generates full-horizon plans of the same lengths.
>
> In robot manipulation and visual maze domains, we adopt the MCTD-Replanning variant introduced in the original MCTD paper, as the MCTD w/o replanning often struggles in these settings due to the complexity of multi-object interactions or the risk of trajectory collapse in long visual rollouts. In this variant, the agent generates a new full-horizon plan at each replanning step (e.g. 500) but executes only the first 50 steps before replanning from the updated state. This process is repeated until the task is completed.
>
> We will clarify these implementation details, including all relevant planning horizons and replanning protocols, in the revised manuscript for completeness.
>
> > The paper would be improved if details such as these were made more explicit in the main text.
>
> We agree with the reviewer’s recommendation to make these details explicit in the main text. In our revision, we will clearly present:
>
> - Definitions of the guidance function and the precise relationship to meta-actions.
> - Exact descriptions of training vs. inference-time trajectory lengths.
> - A clear explanation of replanning strategies, intervals, and their impact on computational efficiency.
>
> We sincerely appreciate reviewer pbF8’s constructive feedback and believe that these revisions will significantly strengthen the quality and self-containedness of our work.

---

> > ### Comment · Reviewer_pbF8 · 2025-08-04
> >
> > Thank you for your thorough response. I now understand all the details I had asked about, and agree that the clarity of the paper will be improved with your proposed changes. Overall, I continue to think this a solid paper with a practical and very effective approach to speed up MCTD.
> >
> > One minor note: I'd like to second the comment by reviewer EsS3 that exploring learned value functions or other sorts of test-time guidance would be an interesting area for future work. The current guidance sets control exploration vs. exploitation at each node, but this seems to be at least somewhat conceptually redundant with the function of the tree search itself.

---

> ### Author Response · Authors · 2025-08-05
>
> We sincerely thank the reviewer for the insightful feedback and the opportunity to improve our manuscript. We have thoroughly revised the manuscript to address the points raised. Regarding the learned value function, we agree that this is a promising direction for future research. We are confident that your valuable comments have significantly strengthened our paper.

---

### Official Review · Reviewer_FWQE · 2025-07-02

**Clarity:** 3
**Significance:** 3
**Originality:** 3
**Rating:** 5
**Confidence:** 3

**Summary:**

The authors propose Fast Monte Carlo Tree Diffusion (Fast-MCTD) to accelerate the planning time of conventional MCTD. They first introduce Parallel MCTD, which enables parallel computation in the expansion and simulation stages to speed up the MCTD process, and implement Redundancy-Aware Selection (RAS) as the selection strategy. Second, they present Sparse MCTD, which trains the diffusion model using sparse, coarse-grained trajectories. Experimental results demonstrate that Fast-MCTD achieves performance comparable to standard MCTD while significantly reducing planning time.

**Questions:**

1. How does Fast-MCTD compare to subgoal-based Diffuser variants that are designed for long-horizon planning, such as the hierarchical approaches in [1] and [2], in terms of both computational cost and performance?

2. How is the Redundancy-Aware Selection (RAS) policy in Eq. (5) derived? Is it adapted from prior work on parallelized MCTS? Were there any challenges in formulating this policy?

3. Are the sparse coarse trajectories used to train the diffusion model in addition to the original trajectories, or do they replace the original trajectories?

[1] Chen, Chang, et al. Simple Hierarchical Planning with Diffusion. The Twelfth International Conference on Learning Representations (ICLR), 2024.

[2] Li, Wenhao, et al. Hierarchical Diffusion for Offline Decision Making. International Conference on Machine Learning (ICML), 2023.

**Ethical Concerns:**

["NO or VERY MINOR ethics concerns only"]

**Final Justification:**

The authors have provided valuable responses in the rebuttal stage that have mostly addressed my concerns. While I believe the idea is not particularly novel and incorporates elements from prior works, the results are excellent and the paper is well written, warranting acceptance.

**Limitations:**

The authors claimed that they discussed the limitations of their work in a separate section, but such a section cannot be found in the paper.

**Paper Formatting Concerns:**

There are no major formatting issues in the paper.

**Quality:**

3

**Strengths And Weaknesses:**

Strengths:

1. The experimental results presented in the paper show a significant improvement in planning speed—up to 100× faster—compared to the standard MCTD method.

2. The paper includes comprehensive experiments to evaluate the effectiveness of the proposed Fast-MCTD approach.

3. Concrete implementation details and extensive experimental configurations are provided in the supplementary materials, enhancing reproducibility.

Weaknesses:

1. The work appears to be an incremental extension of the original MCTD method. The primary contribution lies in applying parallel computation to certain stages of MCTD, which limits the novelty of the paper.

2. According to the original MCTD paper, the number of possible search trajectories is already bounded, meaning that the computational cost of standard MCTD does not grow exponentially without limit. Moreover, the proposed parallelization in Fast-MCTD does not reduce the exponential complexity in the worst case, but rather achieves a practical speedup by a factor of $K$.

---

> ### Author Rebuttal · Authors · 2025-07-31
>
> > The work appears to be an incremental extension of the original MCTD method. The primary contribution lies in applying parallel computation to certain stages of MCTD, which limits the novelty of the paper.
>
> We sincerely thank the reviewer for their valuable feedback. We appreciate this opportunity to clarify the primary contributions of our work.
>
> First of all, we would like to emphasize that our main contribution lies in unleashing the full potential of the MCTD method as a practically adoptable solution by resolving its primary bottleneck—speed—rather than proposing a conceptually novel methodology.
>
> Nevertheless, we respectfully wish to clarify that Fast-MCTD is also not merely a parallel extension of MCTD. Seemingly, it might look like an adoption of the MCTS parallelization. However, to make the adoption effective in the context of diffusion-based planning, we additionally propose **Redundancy-Aware Selection** and **Parallel Denoising**. These are non-trivial additions designed to efficiently navigate the vast search space inherent to diffusion models, a challenge that standard parallelization does not address. Furthermore, our contribution is not limited to parallel search. We also introduce **Sparse-MCTD**, a complementary approach that fundamentally enhances efficiency by enabling coarsen planning.
>
> Together, **these techniques transform MCTD from a proof-of-concept into a practical tool for long-horizon or large-scale planning, achieving up to 100× speedup while retaining performance.** By addressing both diffusion- and search-specific inefficiencies, Fast-MCTD improves the practical usability of standard MCTD for long-horizon planning.
>
> > The number of possible search trajectories is already bounded, meaning that the computational cost of standard MCTD does not grow exponentially without limit.
>
> While it is true that the number of search trajectories in MCTD is theoretically bounded, we emphasize that **practical planning cost still grows exponentially** with the number of subplans $\bar{s}$, which increases with the planning horizon. As detailed in Section 3, the search complexity scales as $O(N_{\text{child}}^{\bar{s}})$, making long-horizon tasks such as Giant PointMaze and visual mazes computationally expensive in practice. If the bound is set too high, computational overhead becomes significant; if set too low, exploration is insufficient and plan quality degrades. Also, in practice, many applications require not only bounded search time but also fast search.
>
> > The proposed parallelization in Fast-MCTD does not reduce the exponential complexity in the worst case, but rather achieves a practical speedup by a factor of $K$.
>
> As you acknowledged, the main contribution of our work is to improve the practical speedup of MCTD. And, we agree that reducing the worst-case exponential complexity is not our main goal. However, the sparse MCTD allows to reduce $\bar{s}$ by planning over temporally coarsened trajectories. This reduces the planning depth exponentially, fundamentally altering the computation complexity—not just the runtime.
>
> $
> C_{\text{S-MCTD}} = \mathcal{O}\left(N_{\text{child}}^{\bar{s} / H} \cdot C_{\text{coarse}}\right)
> $
>
> In standard MCTD, the search complexity scales as:
>
> $
> C_{\text{MCTD}} = \mathcal{O}(N_{\text{child}}^{\bar{s}} \cdot C_{\text{sub}})
> $
>
> > How does Fast-MCTD compare to subgoal-based Diffuser variants that are designed for long-horizon planning, such as the hierarchical approaches, in terms of both computational cost and performance?
>
> We thank the reviewer for this valuable question. While both Fast-MCTD and subgoal-based diffusion planners aim to address long-horizon planning through abstraction, they focus on fundamentally different aspects of the problem.
>
> - **Hierarchical Diffuser (HD)** [1] does **not perform explicit search**, but instead generates trajectories using subgoal conditioning. This approach results in much larger improvements in **training time** than in inference time (e.g., training time reduced from 132.7s to 8.7s, while planning time drops from 9.9s to 3.1s in Med-Maze).
> - In contrast, Fast-MCTD performs tree-based search at inference time. By coarsening trajectories and reducing the overhead on denoising process and the number of subplans from $\bar{s}$ to $\bar{s}/H$, Fast-MCTD achieves an exponential reduction in the search space, leading to significant speedups during planning.
>
> We perform a fair comparison using guidance-based planning for both methods on PointMaze tasks. **The result shows that Fast-MCTD significantly outperforms HD with better efficiency.**
>
> |  |  | Planning Time |  |  | Success Rates |  |
> | --- | --- | --- | --- | --- | --- | --- |
> |  | medium | large | giant | medium | large | giant |
> | Diffuser | 6.6 ± 0.10 | 6.5 ± 0.10 | 6.4 ± 0.1 | 0.58 ± 0.06 | 0.44 ± 0.08 | 0 ± 0 |
> | Hierarchical Diffuser | 4.48 ± 0.14 | 4.57 ± 0.10 | 4.90 ± 0.31 | 0.62 ± 0.06  | 0.24 ± 0.08 | 0.00 ± 0.00 |
> | MCTD | 59.2 ± 27.1 | 174.6 ± 27.2 | 264.2 ± 33.8 | 1 ± 0 | 0.98 ± 0.06 | 1 ± 0 |
> | Fast-MCTD | 3.6 ± 0.6 | 4.1 ± 0.4 | 2.4 ± 0.1 | 1 ± 0 | 0.80 ± 0.00 | 1 ± 0 |
>
> Particularly, Fast-MCTD achieves over 100× inference-time speedup through exponential search space reduction with sparse planning, while Diffuser and Hierarchical Diffuser achieve only modest planning speedup under the same evaluation protocol.
>
> [1] Simple Hierarchical Planning with Diffusion., Chen et al., ICLR`24
>
> > How is the **Redundancy-Aware Selection (RAS) policy in Eq. (5) derived**? Is it adapted from prior work on parallelized MCTS? Were there any challenges in formulating this policy?
>
> We thank the reviewer for the insightful question regarding our Redundancy-Aware Selection (RAS) policy. The reviewer is correct that the design of RAS is inspired by prior work on parallelized MCTS [1, 2], particularly the concept of virtual loss introduced in [2]. Furthermore, we apply a penalty only to the visit counts to encourage a more exploitative search.
>
> Regarding the challenges in formulating this policy, the key consideration was the sensitivity to the hyperparameter $w$. To address this, we conducted an extensive ablation study (presented in Table 6 for the PointMaze-Giant tasks). The results demonstrate that Fast-MCTD exhibits **robust performance across a wide range of w values**, indicating that precise tuning is not critical for achieving strong results.
>
> [1] Parallel monte-carlo tree search. Guillaume et al., In Computers and Games, 2008.
>
> [2] Mastering the game of go with deep neural networks and tree search. Silver et al., Nature 2016.
>
> > **Are the sparse coarse trajectories used to train the diffusion model in addition to the original trajectories, or do they replace the original trajectories?**
>
> We thank the reviewer for the clarifying question. In S-MCTD, **coarsened trajectories replace the original ones** during model training. That is, we train a single diffusion model on uniformly subsampled sequences (e.g., every $H$ steps), instead of using the original fine-grained data.
>
> > The authors claimed that they discussed the limitations of their work in a separate section, but such a section cannot be found in the paper.
>
> We thank the reviewer for their comment. While we discussed key limitations in Section 7 (e.g., reliance on parallelism and new hyperparameters), we acknowledge the absence of a dedicated “Limitations” section may have reduced clarity. We will restructure these points into a clearly marked section in the revision.

---

> > ### Comment · Reviewer_FWQE · 2025-08-04
> >
> > I thank the authors for their detailed responses, which have mostly addressed my concerns. The paper is well written, and I will raise my score.

---

> > > ### Author Response · Authors · 2025-08-05
> > >
> > > We are sincerely grateful to the reviewer for their thoughtful re-evaluation and insightful comments, which were instrumental in enhancing the discussion of our work and strengthening our experimental analysis.

---

### Official Review · Reviewer_hTDe · 2025-07-04

**Clarity:** 3
**Significance:** 4
**Originality:** 2
**Rating:** 5
**Confidence:** 4

**Summary:**

This paper introduces Fast-MCTD, a framework designed to accelerate the performance of Monte Carlo Tree Diffusion (MCTD), a state-of-the-art diffusion-based planner. The authors identify two key bottlenecks in MCTD: the sequential nature of tree search and the high computational cost of iterative denoising over long trajectories. To address these, they propose two primary techniques: Parallel MCTD (P-MCTD), which parallelizes the search process using delayed tree updates and a redundancy-aware selection mechanism, and Sparse MCTD (S-MCTD), which reduces the effective planning horizon by operating on coarsened, subsampled trajectories. Experimental results across several challenging long-horizon tasks demonstrate that Fast-MCTD can achieve significant speedups over the original MCTD, while largely maintaining or sometimes even improving planning success rates.

**Questions:**

* Could you provide a more concrete example of the tree construction? Specifically, how is a trajectory partitioned into subplan nodes, and how are the meta-actions (guidance schedules) applied to these nodes during expansion? It'd be nice if the authors can expand on the details of each procedure of Fast-MCTD in the appendix.

* The S-MCTD method creates abstract trajectories by uniformly subsampling states every H steps. How does this method of abstraction compare to more structured approaches, such as the options framework by Sutton et al. (1999), where temporally-extended actions are typically defined by sub-goals or semantic criteria? Does uniform subsampling risk creating abstract waypoints that are dynamically challenging for the low-level controller to achieve, and could you comment on whether this approach captures semantically meaningful sub-tasks versus serving primarily as an effective computational heuristic?

Reference:
[1] Sutton et al. (1999): Between MDPs and semi-MDPs: A framework for temporal abstraction in reinforcement learning.

**Ethical Concerns:**

["NO or VERY MINOR ethics concerns only"]

**Final Justification:**

This is a strong, high-impact paper, and the authors' rebuttal fully addressed all of my questions.

My initial concerns about the paper's clarity and the justification for its subsampling method have been resolved. The authors committed to adding implementation details to the appendix and provided a compelling, evidence-backed explanation for their design choice.

I have no remaining issues and view this as a clear accept.

**Limitations:**

yes

**Quality:**

3

**Strengths And Weaknesses:**

__Strengths__

* Practical impact: The work delivers a significant and practical contribution by drastically reducing the planning time of a state-of-the-art diffusion planner. The reported 80-110x speedups on challenging long-horizon tasks make the underlying MCTD method far more viable for practical use.

* The authors present a well-designed approach, identifying two distinct inefficiencies (between-rollout and within-rollout) and proposing a specific, well-motivated solution for each (P-MCTD and S-MCTD, respectively).

* Claims are backed by a comprehensive evaluation on diverse benchmark tasks. The ablation studies on key hyperparameters like parallelism degree and interval size are reasonable and provide convincing evidence for the method's effectiveness.

__Weaknesses__

* Lack of self-containedness: The paper relies heavily on prior work to explain core mechanics. Critical details on how the diffusion model is trained for this task, and how trajectories are partitioned into subplans to form the tree structure, are not sufficiently explained. This lack of clarity hinders a full understanding and reproducibility.

---

> ### Author Rebuttal · Authors · 2025-07-31
>
> We thank reviewer hTDe for the thoughtful feedbacks and valuable suggestions.
>
> > Details on how the diffusion model is trained
>
> We appreciate the reviewer’s interest in implementation details. The diffusion model in Fast-MCTD follows the same training protocol as MCTD, and we explicitly restate all relevant configurations in Appendix A.4 for clarity and completeness. We will further revise the appendix to ensure the training procedure is fully transparent and reproducible.
>
> **We follow the same training protocol as prior diffusion planners such as Diffuser and Diffusion Forcing.** Concretely, the model is trained on offline datasets of trajectories collected in OGBench. The training objective follows a standard $\mathbf{x}_0$ prediction loss, where the model learns to recover the original trajectory $\mathbf{x}_0$ from noisy inputs. We employ a Transformer architecture with stacked attention blocks, trained using cosine schedules. All hyperparameters, including learning rates, batch sizes, and sampling schedules, are fully reported in the appendix for reproducibility.
>
> Additionally, we plan to release our full implementation shortly, which will further support reproducibility.
>
>
> > How is a trajectory partitioned into subplan nodes, and how are the meta-actions (guidance schedules) applied to these nodes during expansion?
>
> For planning, we simply partition each trajectory into fixed-length segments of $S$ timesteps. Each segment, or subplan, is used as a high-level node in the search tree. Each node is denoised independently using a blockwise schedule, which allows early subplans to be generated first and later ones to be conditioned on them, creating an autoregressive structure within the diffusion framework.
>
> During tree expansion, meta-actions are defined as **guidance levels** selected from a predefined set (e.g., $\{0, 1, 2, 4\}$), controlling the strength of classifier guidance for each subplan. These levels modulate how strongly each node is biased toward goal achievement, enabling dynamic exploration–exploitation balancing across the tree.
>
> We will expand these details in the appendix.
>
> > The S-MCTD method creates abstract trajectories by uniformly subsampling states every H steps. How does this method of abstraction compare to more structured approaches, such as the options framework by Sutton et al. (1999), where temporally-extended actions are typically defined by sub-goals or semantic criteria?
>
> We thank the reviewer for this insightful question.
>
> S-MCTD adopts a fixed interval subsampling strategy (e.g., every $H$ steps)—rather than defining subgoals through semantic or hierarchical criteria, as done in classical options frameworks. This approach follows recent diffusion planners such as Hierarchical Diffuser [1,2] and PlanDQ [3], and offers a simple yet general abstraction mechanism.
>
> Empirically, this strategy has proven highly competitive: Hierarchical Diffuser [1] reports that fixed-interval subsampling **outperforms adaptive subgoal selection** methods like HDMI [4], suggesting that even simple fixed-interval structures can outperform learned subgoal discovery methods.
>
> [1] Simple Hierarchical Planning with Diffusion., Chen et al., ICLR’24
>
> [2] DiffuserLite: Towards Real-time Diffusion Planning, Dong et al., NeurIPS’24
>
> [3] Hierarchical Plan Orchestration via D-Conductor and Q-Performer., Chen et al., ICML’24
>
> [4] Hierarchical diffusion for offline decision making, Li et al., ICML’23
>
> > Does uniform subsampling risk creating abstract waypoints that are dynamically challenging for the low-level controller to achieve?
>
> We acknowledge that overly coarse waypoints can challenge low-level controllers. In S-MCTD, we address this by conducting an ablation over the interval size $H$, which reveals a clear trade-off: increasing $H$ improves efficiency, but can degrade success rates due to infeasible transitions. Based on this, we select $H$ to balance planning speed with controller compatibility.
>
> > Could you comment on whether this approach captures semantically meaningful sub-tasks versus serving primarily as an effective computational heuristic?
>
> We thank the reviewer for this thoughtful question. **S-MCTD does not attempt to discover semantically meaningful subgoals.** This is a **deliberate design choice**, not a limitation. Semantic subgoal discovery often requires task-specific priors, auxiliary supervision, or added complexity—all of which reduce scalability and generality across domains.
>
> Instead, our method adopts a **simple, task-agnostic abstraction** based on fixed-interval subsampling. While it does not align with semantic task boundaries, it enables robust and tractable long-horizon planning without additional annotation or modeling overhead. Prior works, such as Hierarchical Diffuser [1, 2], have shown that this simple approach actually outperforms the adaptive subgoal method like HDMI [3].
>
> Moreover, this regular structure is especially well-suited for **parallel rollout planning**, as required in P-MCTD. Semantically chunked subgoals could introduce variability and irregularity that would hinder batching and GPU-efficient parallel denoising.
>
> [1] Simple Hierarchical Planning with Diffusion., Chen et al., ICLR’24
>
> [2] DiffuserLite: Towards Real-time Diffusion Planning, Dong et al., NeurIPS’24
>
> [3] Hierarchical Diffusion for Offline Decision Making., Li et al., ICML’23

---

> > ### Comment · Reviewer_hTDe · 2025-08-08
> >
> > Thank you for the detailed and helpful response.
> >
> > Your clarifications on the training protocol and the partitioning of trajectories have addressed my main concern about the paper's self-containedness. I encourage you to incorporate these missing details into the appendix of your revised paper.
> >
> > Your answer regarding uniform subsampling versus more structured approaches was insightful. The reference to prior work showing that fixed-interval subsampling can outperform adaptive subgoal methods is a compelling justification for your design choice. I suggest including this discussion in the revised paper, as it effectively contextualizes your method.

---

> ### Author Response · Authors · 2025-08-08
>
> We appreciate you acknowledging that our rebuttal has resolved your primary concerns. As suggested, we will incorporate a more detailed description of the training protocol and a comparative discussion between uniform subsampling and more structured approaches into the revised manuscript.

---

### Note · Authors · 2025-08-16

Dear Area Chair and Reviewers,

We are sincerely grateful for the insightful and constructive discussion period. We are pleased that our rebuttal successfully addressed the initial questions, as reflected in the **positive follow-up comments from all four reviewers, as well as subsequent score updates**.

This review process has been invaluable for our work. Based on the excellent suggestions, we will incorporate the following key revisions into the final manuscript to significantly enhance its quality and clarity:

- **Enhanced Clarity and Reproducibility:** We will provide more comprehensive details on our methodology, including the full training protocol, the definition of the guidance function, and a clearer explanation of replanning strategies (addressing feedback from **Reviewers hTDe, pbF8**). We will also release our code to guarantee full reproducibility.

- **Sharpened Contribution and Context:** We will refine the discussion to better contextualize our work against prior work. We will also highlight how S-MCTD offers a fundamental complexity reduction, not just a practical speedup, and will further emphasize the distinct innovations of this work (as suggested by **Reviewers hTDe, FWQE, pbF8**).

- **Expanded Discussion and Future Work:** The discussion will be substantially updated to include a comparison between uniform versus structured subsampling, trade-offs of sparse planning, and generalization performance, and we will also add a dedicated limitations section. The future work section will also be expanded to include applications like multi-agent planning and learned value functions (based on suggestions from **Reviewers hTDe, FWQE, pbF8, EsS3**).

We are confident that these revisions will make our paper stronger and more impactful. We believe our work represents a significant step forward in making diffusion-based planning practical and high-performing.

Thank you again for your time and positive consideration of our work.

Sincerely,

The Authors

---

### Decision · Program_Chairs · 2025-09-17

**Decision:**

Accept (spotlight)

**Comment:**

The paper builds on the recently proposed Monte Carlo Tree Diffusion (MCTD), proposing a new variant Fast-MCTD improving its speed and scalability including parallel rollouts via delayed tree updates - a usual feature for MCTS game programs. The length of rollouts are reduced by operating on sparse, coarse-grained trajectories. Empirical evaluation show that the proposed variant improves significantly on MCTD in terms of speed.

While the reviewers argued that the ideas behind the new variant are not novel for MCTS, the proposed variant seems strong enough based on the empirical evaluation presented.

The reviewers were satisfied by the author-reviewer discussions, and felt that the paper is worthy of publication. We advise the authors revise the paper according to the discussions.